In-vitro comparative thermo-chemical aging and penetration analyses of bioactive glass-based dental resin infiltrates

http://orcid.org/0000-0002-7259-5141 Ahmed Syed Zubairuddin 1 szahmed@iau.edu.sa
http://orcid.org/0000-0002-2165-800X Khan Abdul Samad 1
Alshehri Maram 2
Alsebaa Fatimah 2
Almutawah Fadak 2
Mohammed Aljeshi Moayad 2
Tufail Shah Asma 3
Md Sabri Budi Aslinie 4
http://orcid.org/0000-0002-9473-4739 Akhtar Sultan 5
Abu Hassan Mohamed Ibrahim 4 6
1 Department of Restorative Dental Sciences College of Dentistry, Imam Abdulrahman Bin Faisal University , Dammam, Eastern Province , Saudi Arabia
2 Imam Abdulrahman Bin Faisal University, College of Dentistry , Dammam, Eastern Province , Saudi Arabia
3 Interdisciplinary Research Centre in Biomedical Materials (IRCBM), COMSATS University Islamabad , Lahore Campus , Pakistan
4 Faculty of Dentistry, Universiti Teknologi MARA , Sungai Buloh, Slengor , Malaysia
5 Department of Biophysics, Institute for Research and Medical Consultations, Imam Abdulrahman Bin Faisal University, Dammam , Saudi Arabia
6 Faculty of Dentistry, MAHSA University , Jenjarom, Selangor , Malaysia
Mishra Yogendra
Electronic publication date: 2025 Jan 28
Publication date: 2025
Volume: 13
Electronic Location ID: e18831
Received 2024 Apr 19; Accepted 2024 Dec 17
Copyright: © 2025 Ahmed et al.
Copyright year: 2025
Copyright holder: Ahmed et al.
License: This is an open access article distributed under the terms of the Creative Commons Attribution License, which permits unrestricted use, distribution, reproduction and adaptation in any medium and for any purpose provided that it is properly attributed. For attribution, the original author(s), title, publication source (PeerJ) and either DOI or URL of the article must be cited.
License URL: https://creativecommons.org/licenses/by/4.0/

Keywords: White spot lesions, Bioactive glass, Resin infiltrant, Microhardness, Surface roughness, Thermo-cycling, Chemical aging, pH-cycling, Micro-CT, Microleakage

Funding: The authors received no funding for this work.

==============================
Background

Teeth with small to moderate cavities can be repaired with enamel resin infiltrants, a form of dental restorative material. In dental materials, it is standard practice to include several filler particles for experimental use in dental resin infiltrates. The resin’s BG particles penetrate the lesion and release ions that combine with saliva to provide a mineral-rich environment that can strengthen enamel and heal. This study aimed to compare resin infiltrants based on three types of bioactive glass materials and investigate the penetration depth, microleakage, and the effect of thermal and chemical aging.

Methodology

A triethylene glycol dimethacrylate (TEGDMA) and urethane dimethacrylate (UDMA)-based experimental resin infiltrate was prepared. Initial mixing was done manually for 1 h at room temperature, followed by another mix for 30 min on a magnetic stirrer. This prepared resin, called “PURE RESIN” was then further incorporated with three different types of bioactive glasses, i.e., Bioglass (45S5), boron-substituted (B-BG), and fluoride-substituted (F-BG). Initial manual mixing for 1 h, followed by ultrasonic mixing for 3 min and then proceeded for the final mixing on a magnetic stirrer for 24 h in a dark room at ambient temperature. Human-extracted teeth were demineralized, and the experimental resins were infiltrated on the demineralized surface. The surface area, pore size, and volume of the demineralized surface were measured. The microleakage and penetration depth were analyzed with the stereomicroscope and micro-CT, respectively. The samples were challenged with the pH cycle for 14 days, followed by a scanning electron microscope (SEM). Thermocycling (5,000 cycles) and chemical aging (4 weeks) were conducted, followed by microhardness, surface roughness, and SEM analyses. Statistical analyses were conducted after each test.

Results

The F-BG group achieved the highest initial and day 14 penetration coefficients. There was a superior dye penetration with the microleakage analysis in the F-BG group. The 45S5 group had the highest average penetration depth via micro-CT analysis. After thermocycling and chemical aging, the micro-hardness was reduced (non-significantly) among all samples except the F-BG group in post-chemical aging analysis, whereas the surface roughness was significantly increased. SEM images showed the presence of micro-pits on the surfaces after the thermal and chemical aging.

Conclusion

The F-BG group achieved the highest initial and day 14 penetration coefficients. There was a superior dye penetration with the microleakage analysis in the F-BG group. The 45S5 group had the highest average penetration depth via micro-CT analysis. After thermocycling and chemical aging, the micro-hardness was reduced (non-significantly) among all samples except the F-BG group in post-chemical aging analysis, whereas the surface roughness was significantly increased. SEM images showed the presence of micro-pits on the surfaces after the thermal and chemical aging.

Introduction

According to the World Health Organization (2022), almost 3.5 billion people worldwide suffer from oral diseases, with three out of every four individuals residing in middle-income nations. It is reported that nearly 514 million children worldwide suffer from primary tooth decay, while 2 billion adults are thought to have permanent tooth decay. In the primary dentition, the mean prevalence of cavitated carious lesions was 33.85% (Carvalho, Thylstrup & Ekstrand, 1992; Leroy et al., 2009; Al-Samadani & Ahmad, 2012; Agrawal, Bhagat & Shrestha, 2023).

There has been a paradigm shift in managing dental caries lesions in the last decade. These changes have evolved the whole traditional concept of treating dental caries (Innes et al., 2019). From initial carious lesions to large compound or complex lesions, the new concept emphasizes a preventive approach comprising non-invasive or minimally invasive methods (Frencken, 2017). Various approaches have been proposed and implemented to meet minimal invasiveness requirements (Lygidakis et al., 2022). Materials used to treat early caries lesions include dental resin infiltrants for restoring white spot lesions (WSL). ICON (DMG Dental Products UK Ltd, Warrington Cheshire, UK), Advantage Arrest™ (Silver Diamine Fluoride, Elevate Oral Care, West Palm Beach, FL, USA), Clinpro White Varnish (3M Center. St. Paul, MN, USA), Delton Plus (Dentsply, York, PA, USA), Novamin (GSK, Middlesex, UK), and CPP-ACP (Recaldent, Burleigh Heads, Australia), are commercially available materials to treat WSLs.

Dental restorative materials have been reinforced with nanofillers to improve their physico-mechanical properties (Dos Santos et al., 2023; Altaie, 2023). Resin infiltrants are reinforced with bioactive inorganic fillers to enhance their chemo-mechanical characteristics and reduce water sorption and solubility. The effectiveness of the long-term management of WSL can be improved by adding bioactive particles to experimental resin infiltrants (Sfalcin et al., 2017). Since the 1980s, bioactive inorganic materials, including hydroxyapatite, resorbable calcium phosphates, and bioactive glass, have been used in orthopedics and dentistry (Cannillo et al., 2021). Among these, bioactive glasses are a type of bioceramics that can adhere to both hard and soft tissues, may have an antimicrobial effect, and may encourage the development of new tissue (Cannillo et al., 2021; Farag, 2023).

The biological and physiological activities of bioactive glasses (BG) can be tailored with the substitution of ions. An alternative BG composition based on borate (B2O3), known as boron-substituted bioactive glass (B-BG), produces apatite more quickly than silicate-based BG (such as 45S5) (Oluwatosin, El Mabrouk & Bricha, 2023). The bioactivity of BG is significantly improved by fluoride (F) ion replacements (F-BG). When activated by contact with water, the F ions are known to raise the pH of the nearby micro-environment by replacing the calcium (Ca) ions in the glass network (Whatley, 2022; Shah, 2016). F-BG can release fluoride ions, inhibit demineralization, and promote remineralization of the enamel, aiming to improve remineralization capabilities and offering potential benefits in combating the early stages of dental caries. B-BG can release borate ions, which can demonstrate versatile reactivity and contribute to the formation of hydroxyapatite-like structures within the enamel, aiding in the reinforcement and strengthening of the tooth structure. 45S5 can release silicate ions, forming a hydroxyapatite layer on the surface of the enamel and bond with the resin. It exhibits strong bioactivity and facilitates the integration of resin infiltration with the natural tooth structure. Using BGs in resin infiltration can improve the esthetic and functional outcomes of the treatment (Oluwatosin, El Mabrouk & Bricha, 2023) and prevent secondary caries (Ramadoss, Padmanaban & Subramanian, 2022).

Numerous factors can affect the characteristics and potency of dental resin infiltrants. Microleakage can impair the sealing and fortification of the enamel and result in the development of secondary caries (Mariani, Sutrisno & Usman, 2018). Due to variations in lesion depth and enamel structure, deeper penetration may not always be possible but can offer more substantial support for the enamel (Dziaruddin & Zakaria, 2022; Kielbassa et al., 2010). The penetration depth of resin infiltrants affects their ability to remineralize and stabilize tooth structures afflicted by early decay (Ibrahim, Venkiteswaran & Hasmun, 2023). Due to exposure to the oral environment, temperature changes, and acidic challenges, dental resin infiltrants may thermally and chemically age over time. This process can weaken the infiltrate’s bond with the enamel and reduce its ability to prevent future demineralization (Soveral et al., 2021).

By combining microleakage and penetration depth analyses, dental professionals can make informed decisions about the efficacy of different resin infiltrants. They can evaluate which infiltrants provide the best seal to prevent microleakage and penetrate deeper to deliver remineralizing agents effectively. This study aimed to assess the extent of microleakage at the interface between the resin infiltrant and tooth structure, the depth of penetration, and to investigate the hardness, surface roughness, and morphological changes of resin infiltrants based on three different types of BGs when subjected through artificial aging processes. It was hypothesized (H1) that the experimental resin infiltrants would effectively seal the tooth surface and prevent microleakage, penetrate the porous structure of the tooth surface, and the experimental resin infiltrants could resist the artificial aging process to mimic the oral environment conditions. The alternative hypotheses were pitted against the null hypothesis (H0).

Materials and Methods

Synthesis of bioactive glass-based infiltrants

Initially, the ratio of dimethacrylate resins, i.e., triethylene glycol dimethacrylate (Batch # STBH2136; TEGDMA, Sigma-Aldrich Inc., St. Louis, MO, USA) and urethane dimethacrylate (Batch # MKCC7162; UDMA, Sigma-Aldrich Inc., St. Louis, MO, USA) was optimized and set at 75:25, respectively. After calculating the required weight percentages of TEGDMA/UDMA (74.5 wt.% and 24.5 wt.%), the resin matrices were mixed manually for 1 h at room temperature. Afterward, the photoinitiators, i.e., camphorquinone (Batch # MKBX1335V; CQ, Sigma-Aldrich, Inc., St. Louis, MO, USA) 0.5 wt.% and ethyl 4-(dimethylamino) benzoate (Batch # MKBX1335V; EDBA, Sigma-Aldrich, Inc., St. Louis, MO, USA) 0.5 wt.% were also added into the earlier mixed resin in the resin matrices and allowed to mix for 30 min on a magnetic stirrer. All mixing procedures were done at room temperature under a dark environment to prevent exposure to light and to secure premature polymerization. The prepared mixture was labeled as a PURE RESIN (PR) and used as a control group (as it has no fillers in its composition). Three different types of bioactive glasses, i.e., 45S5, B-BG, and F-BG, were mixed to prepare experimental infiltrants. The composition of this commercial bioactive glass “45S5” (G018-144; Schott Glass AG, Mainz, Germany) is silica (SiO2) 43–47%, calcium oxide (CaO) 22.5–26.5%, phosphorus pentoxide (P2O5) 5–7%, and sodium oxide (Na2O) 22.5–26.5%. F-BG and B-BG were prepared by using the ultrasonic precipitation method. The composition of F-BG is SiO2.CaO. Na2O.P2O5.NaF, whereas B-BG is SiO2.CaO. xB2O3.P2O5. The size of the particles of each bioactive glass material was analyzed ((borosilicate BG) ~ 60 nm; (fluoridated BG) ~ 10 nm; and (Bioglass 45S5) ~ 500 nm) using transmission electron microscopy (TEM) (Khan et al., 2023). The density was calculated using the Archimedes Principle, and calculated values for 45S5, F-BG, and B-BG were 2.7, 2.5, and 2.29 g/cm−3, respectively. Initially, the 45S5, F-BG, and B-BG were mixed separately with the PR for 24 h on a magnetic stirrer, whereby the concentration (2.5 wt.%) was optimized. Four experimental infiltrants were made and stored in the dark to avoid early polymerization. The volume of the dental fillers was calculated as described in the literature (Mirică et al., 2020; Kim, Ong & Okuno, 2002):

(1) Fractionoffillervol.%=(wfdfwfdf+wrdr)×100

where wf is the wt. of filler, wr is the weight of resin, df is the density of the filler, and dr is the density of the resin.

The density of each bioactive glass was calculated, and the volume was calculated using V = m/d. As per the calculation done by the formula mentioned above, 45S5 has 1.027 Vol. %, B-BG has 1.116 Vol. %, and F-BG has 1.20 Vol. % respectively.

Sample size calculation

The sample size calculation was done using the mean and standard deviation values from a previous study (Sfalcin et al., 2017). The confidence interval was 95% and 5% margin of error (α), the estimated sample size based on the difference of experimental groups, n = 3 per group, (using the following formula):

(2) n=σ2×(Z1−α/2+Z1−β)2/(μ0−μ1)2.

The power analysis revealed that the sample size was not too large, and a sample size is required to evaluate the effect. Therefore, the investigator decided to use a convenient sampling plan.

Group distribution

The analyses were conducted separately. The samples used for each analysis are given below: pH cycling—five samples per group/analysis (hardness, surface roughness, scanning electron microscopy (SEM) analysis)

Thermocycling—five samples per group/analysis (hardness, surface roughness, SEM analysis)

Chemical immersion—five samples per group/analysis (hardness, surface roughness, SEM analysis)

Preparation of samples

Human-extracted caries-free premolar teeth were collected after obtaining ethical permission from the Scientific Research Ethics Committee, Imam Abdulrahman Bin Faisal University IRB-2022-02-155 & IRB-2023-02-280. Teeth that were extracted due to periodontal and orthodontic reasons were selected. The extracted teeth were disinfected with a 70/30 ethanol solution for 10 min and then stored in a 0.5% thymol solution at a refrigerated temperature. The obtained teeth were then encased in acrylic resin so only their crowns were visible. The teeth were then sectioned in a transverse direction from the cement-o-enamel junction using a diamond-tipped circular saw (Buehler Isomet® 300; Lake Bluff, IL, USA).

Production of white spot lesions

The buccal surfaces of teeth were demineralized, where a 4 mm × 4 mm window was created on the buccal side of the enamel surfaces, and the remaining area was covered with a solitary coat of acid-resistant nail polish (Maybelline, Manhattan, NY, USA). Other surfaces were not processed for demineralization. The samples were immersed in a demineralization solution for 96 h at 37 °C for the demineralization of the open window surfaces to artificially produce white spot lesions (WSL). The demineralization solution was prepared by using 0.05 mM acetic acid solution (lot # J3530; Honeywell, Offenbach, Germany), 2.2 mM CaCl2 (lot # 5119; Cepham Life Sciences, Fulton, MD, USA), and 2.2 mM Na3PO4 (lot. # 22141-27948; Research Products Int., Mount Prospect, IL, USA), and a few drops of KOH (lot # BCCD2228; Sigma Aldrich, St. Louis, MO, USA) were adjusted, and the pH was finally corrected to 4.4.

Application of resin infiltration on WSLs

Then, each experimental resin infiltrant was applied on the enamel surface individually from the application of ICON ETCH® (for 60 s) till the second layer of resin infiltration for 60 s (followed by the manufacturer of the commercial resin material (Ibrahim et al., 2023)) and then exposed to high-intensity blue visible light (650–850 mW/cm2, 420–500 nm, Woodpecker, Guangxi, China) for 40 s to complete the curing process (as per the recommendation of the manufacturer). The method of preparation of WSL and the scanning electron microscopic image of the demineralized window is shown in Fig. 1. Figure 1A shows a window preparation on the tooth surface for the creation of white spot lesions, Fig. 1B scanning electron microscopic view of the prepared surface after the creation of artificial white spot lesion.

Figure 1 Window preparation on tooth surface for the (A) creation of white spot lesions, (B) scanning electron microscopic view of the prepared surface after creation of artificial white spot lesion.

Preparation of samples for Brunauer-Emmett-Teller analysis

The teeth samples were demineralized using the same method mentioned above; however, there was no application of experimental resin material on the surfaces of the teeth as per the requirement of the characterization. The demineralized and normal (untouched) teeth samples were subjected to degassing for 16 h at 40 °C before the analysis and then placed in a sample tube, and contaminants from their surfaces were removed using either a vacuum or a flowing gas. A 2420 Accelerated Area and Porosimeter System were used to examine the sample tube (Micromeritics, Norcross, GA, USA). The analysis port was used to place the sample tube in the system. A 120 °K measurement of the krypton adsorption isotherm was made. The specific surface area, pore volume, and pore size of each group were determined using the standard Brunauer-Emmett-Teller (BET) method.

pH-cycling

The samples prepared from each group were challenged with the pH cycle. A dynamic demineralization/remineralization cycling protocol was carried out to imitate the changes in pH in the mouth. This protocol involved immersing the samples in a demineralized solution for 6 h, followed by 18 h in a remineralized solution. The remineralization solution was prepared with 100 mL of deionized water containing 1.5 mM CaCl2 0.15 mM Na3PO4 (lot. # 22141-27948; Research Products Int., Mount Prospect, IL, USA). The pH of the solution was adjusted to 7 with a few drops of KCl (lot # 0001355892; PanReac AppliChem, Chicago, IL, USA). Whereas the demineralization solution was prepared as mentioned above. The sample was then placed in an incubator at 37 °C. This pH-cycle was repeated daily for 14 days, with the solutions being changed every 72 h to keep the pH and the efficacy of the solution like day 1.

Penetration coefficient of resin infiltrants

The samples prepared for scanning electron microscope (SEM, FEI Inspect, Eindhoven, The Netherlands) analysis followed the same protocol as for other characterization, and the images were viewed and analyzed using ImageJ software. Depth of penetration (DOP) was given a scoring system; score 0 = no penetration; score 1 = intermediate penetration (<25 μm); and score 2 = deep penetration (≥25 μm) (Signore et al., 2021).

Micro-computed tomographic (µ-CT) analysis

Following the same methodology for the preparation of samples, Bruker Skyscan 1172 (Billerica, MA, USA) machine was used to analyze the penetration of each resin infiltrant on the enamel surfaces. The machine was set at <1 μm low contrast resolution (10% MTF): 5 μm Pixel size at maximum magnification: 0.7 to 25 μm to capture the images.

Microleakage analysis

The enamel surface preparation and application of each resin infiltrant were conducted as described above. Then, the samples were immersed for 24 h in a 0.5% solution of basic fuchsin dye (Merck, Darmstadt, Germany) @ 37 °C in an incubator. The samples were removed and cleaned with deionized water and then sectioned buccolingually to observe the microleakage of the resin materials.

Two proficient examiners blinded to the treatment group assignments evaluated microleakage through an ×20 and ×40 stereomicroscope (Olympus Co., Tokyo, Japan). Each examiner independently assessed the stain depth and assigned a score. In cases where disparate scores were reported by the examiners, additional readings were undertaken until a consensus was reached. The scoring for dye penetration utilized a rating system ranging from 0 to 4: 0 = no penetration of methylene blue; 1 = methylene blue penetrates the outer half of enamel; 2 = methylene blue penetrates the inner half of enamel; 3 = methylene blue penetrates the outer half of dentin; 4 = methylene blue penetrates the inner half of dentin (Klaisiri et al., 2020). The fraction of microleakage was estimated by dividing the total distance of penetration of dye (in mm) in contact with the enamel sealant for its whole length (in mm).

Thermocycling of samples

The samples from each group were thermocycled, where thermocycling (SD Mechatronik, Feldkirchen, Germany) was carried out for 5,000 cycles between 5 °C and 55 °C with a dwell time of 15 s. The specimens were cycled continuously to avoid any differences from the samples. After thermocycling, Vicker’s micro-hardness, surface roughness, and morphological analyses (SEM) were conducted for each group. The specifications are mentioned in the subsequent sections.

Chemical immersion of samples

The samples were submerged separately in 1 mL of 75% ethanol solution (PanReac Applichem GmbH, Darmstadt, Germany) for 3 weeks at 37 °C. The surface-to-volume ratio was computed to ensure that each sample was completely immersed in the solution and received the entire demineralization effect. The pH was monitored daily, and the ethanol solution was changed every seven days to maintain the same impact as on day 1. Immediately after the immersion, the samples were taken out from the ethanol solution, followed by the micro-hardness, surface roughness, and morphological analyses.

Vicker’s micro-hardness analysis

The samples were analyzed employing a micro-hardness (MicroMet 6040 Microhardness Testing Machine, Buehler Lake Bluff, IL, USA) with Vicker’s indenter by applying a load of 200 g with 20 s dwell time. All samples were examined both before and after being subjected to thermocycling and chemical aging procedures. Vicker’s number (HV) is determined using the following formula:

(3) HV=1.8544(F/d2)

where HV is the Vickers Pyramid value, F is load and d is the diagonal length in this equation.

Surface roughness analysis

The specimens were tested using a surface roughness test (Contour GT Surface Roughness Tester, Bruker Daltonics GmbH, Bremen, Germany). The average surface roughness (Ra) was used to collect data from pre-thermocycling and chemical immersion into the solution and post-thermocycling and chemical immersion.

Morphological analysis

The morphological pattern of samples from each group was examined with a scanning electron microscope (SEM, JEOL Ltd., TOKYO, Japan; FEI Inspect, Eindhoven, The Netherlands). The specimens were gold sputtered in a vacuum appliance (BalzersSCD 050 Sputter Coater; Liechtenstein, Germany) for 90 s. The scans were collected at various magnifications (×200, ×500, ×1K, ×2K, and ×5K), where the voltage was 5–20 kV.

Statistical analysis

SPSS-20.0 (IBM product, Armonk, NY, USA) was used to carry out statistical analysis. One-way analysis of variance (ANOVA) was carried out to compare distinct groups. The independent t-test was used to compare two groups, the Mann Whitney-U test was used to compare variables with atypical distributions, and the Tukey multiple comparison tests were used to analyze subgroups. The results were assessed with a 95% confidence interval at a significant threshold of p < 0.05.

Results

Breuneur-Emmett Teller analysis

Table 1 shows the BET surface area, pore volume, and pore size of the intact and demineralized enamel surfaces. There was a substantial increase (27.53%) in the surface area from the intact group to the demineralized group. Total pore volume also increased by 26.66%. The pore radius, however, remained almost the same throughout the demineralization process.

Table 1 Textural properties of teeth with observed values of tooth pore-size.

Textural properties	Teeth-A (Non-demineralized)	Teeth-B (Demineralized)	
Surface area (m2/g)	6.61	8.43	
Pore radius (nm)	2.52	2.24	
Pore volume (cm3/g)	0.0045	0.0057	

Penetration coefficient

The SEM images (Figs. 2A–2D) show the cross-sectional area before placing the samples for the pH challenge. It was observed that the experimental infiltrants penetrated the enamel, where the maximum penetration at day 0 was achieved with the F-BG group (25.5 ± 1.5 μm), followed by 45S5 (23.7 ± 2.1 μm), PR (20.32 ± 2 μm), and B-BG group (14.1 ± 2.2 μm) resin groups. The scoring pattern showed that both F-BG and 45S5 resins had deep penetration (>25 μm), while B-BG and PR exhibited intermediate penetration (<25 μm). On day 14, the transverse section showed an almost smooth surface; however, the F-BG group showed minor pores on the surface.

Figure 2 SEM images show the penetration depth of (A-0) PR, (B-0) 4S5, (C-0) B-BG, and (D-0) F-BG resin infiltrant groups at day 0.

After 14 days of pH cycle, the surface changes are observed on (A-14) PR, (B-14) 45S5, (C-14) B-BG, and (D-14) F-BG groups (45S5, Bioglass; B-BG, borate-based bioactive glass; F-BG, fluoride-based bioactive glass; PR, pure resin).

Micro-CT/penetration depth analysis

According to the findings, the 45S5 group had the highest average penetration depth (Table 2) value of 8.42 ± 0.20 µm, followed by the PR group (5.67 ± 0.75 µm), F-BG (5.06 ± 0.97 µm), whereby the B-BG group had the lowest value of 3.75 ± 0.52 µm (Fig. 3).

Table 2 Analytic mean and standard deviation microleakage values of all the resin-infiltrated teeth after immersion into the methylene blue dye.

Group	Microleakage (µm)	Penetration depth (µm)	
45S5	7.94 ± 0.97a	8.42 ± 0.20a,b,c,d	
B-BG	8.35 ± 1.52b	3.75 ± 0.52a	
F-BG	7.48 ± 0.84c	5.06 ± 0.97b,e	
PR	2.84 ± 0.72abc	5.67 ± 0.75d,e	
Sig.	0.0001*	0.0001*	
Notes:

* Shows the significant effect of microleakage and penetration depth between the groups at p < 0.05.

Small letters “abc” show statistically significant differences between the materials vertically.

Small letters “abcde”, show statistically significant differences vertically between the materials (45S5, Bioglass; B-BG, borate-based bioactive glass; F-BG, fluoride-based bioactive glass; PR, pure resin).

Figure 3 Micro-computed tomography images of (A) 45S5, (B) B-BG, (C) F-BG, (D) PR resin infiltrants showing penetration depth (arrows showing) on teeth surfaces (in microns) i.e., 8.42, 3.75, 5.06, and 5.67, respectively.

45S5, Bioglass; B-BG, borate-based bioactive glass; F-BG, fluoride-based bioactive glass; PR, pure resin.

Microleakage

The stereomicroscopic images in Figs. 4A–4E display the findings of the dye penetration (Table 2). The B-BG group registered the highest dye penetration value (8.35 ± 1.52 µm) among all the groups tested, followed by 45S5 (7.94 ± 0.97 µm), F-BG (7.48 ± 0.84 µm). In contrast, the PR group had the lowest value of all the groups (with an average value of 2.84 ± 0.72 µm).

Figure 4 Stereomicroscope images of teeth surfaces with resin infiltrant groups i.e., (A) 45S5, (B) B-BG, (C) F-BG, (D) PR after immersing into the methylene blue dye.

45S5, Bioglass; B-BG, borate-based bioactive glass; F-BG, fluoride-based bioactive glass; PR, pure resin.

Thermocycling-microhardness

Table 3 displays the mean and standard deviation of micro-hardness values before and after thermocycling. In the pre-thermocycling phase, the TOOTH group demonstrated the highest values (168.53 ± 37.97), while the 45S5 group exhibited the lowest values (135.63 ± 12.83). Similarly, in the post-thermocycling analysis, the TOOTH group again showed the highest values (155.63 ± 25.53), while the 45S5 group exhibited the lowest values (132.73 ± 13.97). Neither of the resin groups statistically showed significant values between the groups, not within the groups at the pre- or post-thermocycling stage.

Table 3 Analytic mean and standard deviation microhardness values of resin infiltrant (before and after thermocycling).

Groups	45S5	B-BG	F-BG	PR	TOOTH	p-value	
Pre-thermocycling	135.63 ± 12.83	148.17 ± 17.26	145.30 ± 19.14	159.00 ± 25.48	168.53 ± 37.97	0.595	
Post-thermocycling	132.73 ± 13.97	137.80 ± 11.30	137.63 ± 14.25	147.83 ± 19.98	155.63 ± 25.53	0.622	
Sig.	0.063	0.141	0.201	0.131	0.219		

Thermocycling-surface roughness

Table 4 and Fig. 5 shows the results of teeth surface roughness after thermocycling. In the pre-thermocycling analysis, the TOOTH group had the highest values (0.78 ± 0.006 µm), while the F-BG group had the lowest values (0.57 ± 0.01 µm) among all the groups. In contrast, in the post-thermocycling analysis, the PR group exhibited the highest values (0.93 ± 0.03) while the F-BG group had the lowest values (0.87 ± 0.002 µm) among all the groups. 45S5 and F-BG showed statistically significant values (p < 0.001), followed by the TOOTH group (p < 0.002), B-BG (p < 0.013), and PR (p < 0.022) groups, respectively. When comparing values between the groups, both pre-thermocycling and post-thermocycling analyses exhibited statistically significant values (p < 0.001).

Table 4 Analytic mean and standard deviation surface roughness (Ra) values (µm) of groups (pre and post thermocycling).

Groups	45S5	B-BG	F-BG	PR	TOOTH	p-value	
Pre-thermo
cycling	0.67 ±
0.009	0.66 ±
0.04	0.57 ±
0.01	0.76 ±
0.03	0.78 ±
0.006	<0.001	
Post-thermo
cycling	0.91 ±
0.008	0.91 ±
0.008	0.87 ±
0.002	0.93 ±
0.03	0.90 ±
0.005	<0.001	
Sig.	<0.001*	<0.013*	<0.001*	<0.022*	<0.002*		
Note:

* Shows the significant effect of pre vs. post solution on micro-hardness within each group at p < 0.05.

Figure 5 After thermal aging of 5,000 cycles, the surface roughness view of (A) 45S5, (B) B-BG, (C) F-BG, (D) PR, and (E) TOOTH only groups.

45S5, Bioglass; B-BG, borate-based bioactive glass; F-BG, fluoride-based bioactive glass; PR, pure resin.

Thermocycling-SEM

The morphological pattern presented a few micro-cracks on 45S5 (Fig. 6A) resin infiltrant material after the thermocycling process; however, the surface was smooth, and no pits were observed. Similar findings were observed for the B-BG (Fig. 6B) and F-BG (Fig. 6C) groups, whereas for the PR (Fig. 6D) and TOOTH (Fig. 6E) groups, the samples lost smoothness, and tiny pits were observed. No sign of any fracture or cleavage was observed.

Figure 6 SEM images of the (A) 45S5, (B) B-BG, (C) F-BG, (D) PR, and (E) TOOTH only groups after thermocycling procedure.

Minor cracks and fissures (arrows showing) are observed after 5,000 cycles. 45S5, Bioglass; B-BG, borate-based bioactive glass; F-BG, fluoride-based bioactive glass; PR, pure resin.

Chemical immersion-microhardness

Table 5 shows the mean and standard deviations of surface micro-hardness for all enamel surfaces before and after chemical immersion of 3 weeks. The TOOTH group had the highest pre-chemical aging value (182.0 ± 5.98), whereas the PR group had the lowest values (125.9 ± 0.45). In comparison, after 4 weeks of chemical immersion, the TOOTH group had the highest micro-hardness values (168.86 ± 4.32), while the PR group exhibited the lowest values (123.58 ± 2.20). Statistical significance (p < 0.003) was observed only in the F-BG resin infiltrant groups, while no other resin groups demonstrated any significant differences. However, within the groups, all the groups showed statistically significant values observed both in pre-chemical aging (p < 0.001) and post-chemical aging (p < 0.001).

Table 5 Analytic mean and standard deviation microhardness values of resin infiltrant (pre- and post-chemical immersion).

Groups	45S5	B-BG	F-BG	PR	TOOTH	p-value	
Pre-immersion	139.95 ± 3.74	153.15 ± 4.02	138.95 ± 2.06	125.9 ± 0.45	182.0 ± 5.98	0.001	
Post immersion	134.65 ± 1.31	148.41 ± 0.86	134.1 ± 1.81	123.58 ± 2.20	168.86 ± 4.32	0.001	
Sig.	0.148	0.123	0.003*	0.155	0.069		
Note:

* Shows the significant effect of pre vs. post solution on micro-hardness within each group at p < 0.05.

Chemical immersion-surface roughness

Table 6 and Fig. 7 represents the roughness values for the teeth after the chemical aging procedure. When assessing values, the TOOTH group exhibited the highest values (0.70 ± 0.006 µm), and the PR group showed the lowest values (0.58 ± 0.01 µm) in pre-chemical aging. In contrast, the F-BG group displayed the highest values (0.97 ± 0.01 µm), and the PR group presented the lowest values (0.86 ± 0.01 µm) in post-chemical aging.

Table 6 Analytic mean and standard deviation surface roughness values (µm) of resin infiltrant (before and after chemical aging).

Groups	45S5	B-BG	F-BG	PR	TOOTH	p-value	
Pre-chemical	0.63 ± 0.01	0.68 ± 0.09	0.65 ± 0.01	0.58 ± 0.01	0.70 ± 0.006	0.001	
Post-chemical	0.96 ± 0.13	0.95 ± 0.18	0.97 ± 0.01	0.86 ± 0.01	0.87 ± 0.01	0.001	
Sig.	0.001*	0.003*	0.001*	0.003*	0.002*		
Note:

* Shows the significant effect of pre vs. post solution on micro-hardness within each group at p < 0.05.

Figure 7 After chemical aging of 4 weeks, the surface roughness view of (A) 45S5, (B) B-BG, (C) F-BG, (D) PR, and (E) TOOTH only groups.

45S5, Bioglass; B-BG, borate-based bioactive glass; F-BG, fluoride-based bioactive glass; PR, pure resin.

All the groups demonstrated statistically significant values in response to chemical immersion, within the groups with 45S5 (p < 0.001), B-BG (p < 0.003), F-BG (p < 0.001), PR (p < 0.003), and TOOTH group (p < 0.002). In contrast, statistically significant values (p < 0.001) were observed between the groups before and after immersion into the chemical aging solution.

Chemical immersion-SEM

SEM images showed (Figs. 8A–8E) roughness after chemical aging immersion. No cracks and micro-cavities were observed on the surface of the 45S5 group (Fig. 8A). B-BG group (Fig. 8B) showed the same trend, whereas some irregular surfaces with cauliflower bunch formation were found. The F-BG (Fig. 8C) surfaces showed minor pitting effects and accumulated resin materials in bunches within the matrix. PR (Fig. 8D) showed some pitting on its surface, whereas the resin matrix was not affected much and showed smoothness. The tooth group (Fig. 8E) was not affected much after chemical immersion.

Figure 8 SEM images of the (A) 45S5, (B) B-BG, (C) F-BG, (D) PR, and (E) TOOTH only groups after chemical aging.

Minor cracks and fissures (arrows showing) are observed after 4 weeks of immersion in the chemical solution. 45S5, Bioglass; B-BG, borate-based bioactive glass; F-BG, fluoride-based bioactive glass; PR, pure resin.

Discussion

A less invasive treatment for early enamel carious lesions is resin infiltration in the pores. It entails using a resin infiltrant substance to penetrate the porous enamel surface and inhibit the progression of caries (Mazzitelli et al., 2022). Bioactive inorganic materials can be added to increase the therapeutic effectiveness of resin infiltrants, subsequently improving the physio-mechanical properties of the tooth surface (Khan et al., 2023). These substances can interact with biological tissues to encourage healing and regeneration. These components can be used in resin infiltrants to improve clinical outcomes with little or no invasiveness (Özcan, Garcia & Volpato, 2021). In this study, three types of bioactive glasses were incorporated into the resin system, and a comparative evaluation was conducted. The penetration depth, microleakage, micro-hardness, surface roughness, and morphological changes were investigated. Overall, the objective has been achieved. The null hypothesis has been rejected as the results showed statistically significant values between the experimental materials.

The results of the present study showed that the demineralization process could increase the enamel pore size volume and make the surface irregular. The increased enamel pores can offer more surface area for bonding (Wu et al., 2022). These densities of the fillers/particles play vital roles in the wetting effect that facilitates the penetration of resin infiltration material (Bonn et al., 2009). The filler with the biggest size and density showed the most change in its physical behavior, as the filler percentage increased when the densities and sizes of the filler particles were taken into consideration. Conversely, the filler that changed the least was the one with the smallest size and lowest density (Emami, Sjödahl & Söderholm, 2005). The incorporation of the fillers into the experimental resin is mentioned and described in past studies (Yun et al., 2022). This property justifies the highest penetration ability of 45S5 compared to others, as it has readily been distributed and absorbed into the resin matrix. The etching process increased the surface area of enamel by approximately 27%, which can promote the insertion of resin material into the tooth pores. During the demineralization of enamel, acid can lead to the gradual dissolution and breakdown of enamel crystallites, causing significant damage to its structure (Lutovac et al., 2017). Similar findings were observed in this study, where the pore radius was diminished to almost 11% due to an etching reaction on enamel. It is assumed that this process may result in the gradual collapse of enamel structures, which can ultimately lead to the obliteration of the previously formed enamel pores. Therefore, restoring the demineralized pores as quickly as possible is essential. Restoration with simple resin infiltrants is insufficient, as it can only inhibit further decline. However, it does not contribute to the remineralization process. The presence of bioactive inorganic particles can help in this regard (Almulhim et al., 2022).

The efficacy of a restoration can be assessed by the ability of a resin infiltrant or monomer to diminish microleakage, which refers to the flow of fluids or bacteria between the tooth and the restoration (Klaisiri et al., 2020). However, the lack of a standard laboratory method to assess resin infiltrant microleakage makes it challenging to compare our findings with those of other investigations. The findings of this study showed that the minimum microleakage was observed in the B-BG group. In contrast, the rest of the groups showed almost the same values without statistically significant differences. Interestingly, the SEM and micro-CT images showed less penetration of the B-BG group than others. It is assumed that it could be due to the nanosize of B-BG, which might have affected the penetration of the resin. This process might reduce the penetration of the resin and reduce microleakage. The presence of the B-BG nanoparticles in the pores could help in remineralization. The micro-CT showed maximum penetration depth with the 45S5 group compared to other groups, whereas SEM images showed more with the F-BG group, however, the difference was non-significant. The high penetration depth could be due to the large particle size of the 45S5 and nanoclusters of the F-BG, which allowed them to remain on the pores’ outer surface and only the resin penetrated.

According to the findings of the present study, the infiltrant was able to infiltrate the simulated WSL and produce a consistent resin layer to inhibit lesion expansion. This could be due to the resin’s improved ability to penetrate (Boruziniat et al., 2021). The penetration depth varied; however, this variation may still matter clinically for remineralizations. Studies have demonstrated that a low-viscosity resin infiltrant, based on TEGDMA, is more effective in masking and penetrating WSLs than adhesives (de Cerqueira et al., 2023; Puleio et al., 2021). Conventionally, only TEGDMA is used in resin infiltrants. TEGDMA is a low-molecular-weight monomer and tends to shrink more than other monomers (Pfeifer et al., 2011). Therefore, UDMA was incorporated with TEGDMA to reduce the chances of shrinkage, though the polymerization shrinkage assessment was beyond the scope of the current study. Bioactive glass particles can encapsulate the enamel crystallites and release Ca, P, Si, and doped ions like F and B, which may enhance the remineralization process. The type of substrate has an impact on the infiltrants’ capacity for penetration. The rate at which liquid permeates a specific capillary bed depends on the penetration coefficient (Saluja, Pradeep & Shetty, 2022). It has been established that relatively large particles did not slow the infiltrant flow, allowing capillary force to drive them into the demineralized enamel (Radwan, 2023). Similar findings were observed in the present study, where the 45S5 group showed better penetration. It was found that the pH cycle did not negatively impact the penetrated resin, whereby more depth was observed. This could be due to the exposed surface; more pores were visible after the pH cycle. The 45S5 group exhibited the most penetration coefficient on day 1.

Thermocycling and chemical aging have impacted the resin surface; however, the experimental resin infiltrants resisted well. When exposed to the complex oral environment, infiltrating resin unavoidably experiences issues such as rapid aging. Thermocycling simulates temperature variations and may result in the resin infiltrate’s expansion and contraction, which could produce microcracks and delamination (George & Nair, 2020). Following the artificial aging process (thermocycling and chemical aging), surface microhardness and roughness were chosen as evaluation parameters because changes in these properties may cause additional issues like staining and discoloration of surfaces, plaque buildup, and subsequent secondary caries formation, or even failure of the resin restoration (Saccucci et al., 2022).

The data revealed reduced microhardness results following thermocycling and chemical aging. Since the TOOTH group exhibited the highest microhardness values, while the 45S5 showed the lowest microhardness values following post-thermocycling amongst all the groups, however, the difference was non-significant. The reduction could be due to temperature fluctuations that could cause tension inside the infiltrant, potentially resulting in mechanical damage (Arslan et al., 2018). Another reason for the reduced values was that the thermocycling could induce microcracks or flaws in the infiltrant, jeopardizing its integrity and lowering its mechanical strength (Venditelli, 2023). Furthermore, repetitive thermal expansion and contraction could compromise the binding between the infiltrant and the tooth surface, resulting in diminished adhesion and possible microleakage (Desai, Stewart & Finer, 2021). However, this study observed non-significant differences, possibly due to bioactive materials that reinforced the polymeric network. Secondly, it is expected that the presence of UDMA might enhance the resistance. The SEM images showed micro-cracks and minor pits, which could be due to the presence of hydrophilic resin, i.e., TEGDMA. Water incorporation into the resin matrix can lead to internal strain, resulting in microscopic fractures (Drummond, 2008; Xie et al., 2022).

The surface roughness increased with both thermal and chemical aging, which is in accordance with previous studies (Saccucci et al., 2022; Alshail & AlSaffan, 2021; Youssef et al., 2020). The samples after thermal aging showed micro-cracks and pits in the PR group. It is assumed that thermal stress could cause a change in the surface quality of areas filled with resin. This can result in the formation of micro-fissures and microcracks. Oral fluids, acidic or alkaline chemicals, and enzymatic degradation can all influence the infiltrate’s characteristics (Ionescu et al., 2022). Bioactive glass-doped infiltrants’ chemical composition might react with the oral environment, changing the release of bioactive ions and potentially altering the material’s bioactivity (Dai et al., 2022).

Chemical aging can also cause surface roughness and mechanical property degradation over time (Hamza et al., 2017). It happens when the ester bonds in the resin infiltrant are hydrolyzed, which may cause a decline in the mechanical properties of the resin. The resin infiltrant may deteriorate due to a drop in pH, which can reduce its mechanical qualities (Mazzitelli et al., 2022). Ethanol storage can soften the polymeric network as well as assess cross-linking density and three-dimensional network development (Alshali et al., 2015; Moraes et al., 2007). The results of this study reacted with the same behavior, affecting the hardness as well as the surface roughness properties. However, with the resin infiltrant on its surface, the teeth exhibited more resistance against the chemical immersion. It was interesting to find the precipitated particles on the surface of the B-BG and F-BG groups compared to the 45S5 and PR. There were a few scattered particles on the 45S5. The presence of new precipitated particles on the B-BG and F-BG could be due to the ion-nucleation theory, where the calcium and phosphate precipitated on the surfaces. This behavior is expected to help in the remineralization process of demineralized tooth surfaces.

Limitations of study and future recommendations

The current research has been conducted only under laboratory conditions, therefore, further clinical studies are necessary to evaluate long-term effectiveness. In-vitro formation of white spot lesions, storage conditions, and simulated “daily” stresses might behave differently from the in-vivo conditions.

Conducting a long-term clinical trial to evaluate the durability of the resin material under different conditions, including biocompatibility, physical properties, improved bonding, and esthetics, is recommended.

Conclusions

Within the limitation of this study, it is concluded that the findings indicated that the 45S5 group displayed better penetration capability and a similar penetration coefficient. Furthermore, B-BG exhibited superior sealing capability. Following thermocycling, the F-BG group revealed the lowest surface roughness and highest microhardness scores. The B-BG demonstrated the maximum resistance to chemical aging, although PR had the smoothest surface. Out of all the experimental groups, the B-BG group showed the best resistance to chemical aging, the least number of changes during thermal analysis, and the highest hardness overall.

Supplemental Information

Supplemental Information 1 All the steps performed throughout the study with the concern characterizations.

Supplemental Information 2 ICON resin infiltrant (commercially available material).

Most of the characterizations were performed along with positive control (ICON resin infiltrant) however, in some of them, ICON was not utilized.

Supplemental Information 3 Composition of Resin infiltrant along with filler particle.

Supplemental Information 4 Analytic Microhardness and Surface Roughness values of All the Resin Infiltrated teeth samples with Mean and Standard Deviation after immersing into the Chemical aging solution and Thermocycling process.

The last two columns show the Microleakage and Penetration depth values for ICON.

Supplemental Information 5 Resin Infiltrant material teeth surface with the process of thermocycling.

ICON resin, baseline scans before thermocycling and after thermocycling.

Supplemental Information 6 Resin Infiltrant material on teeth into the Chemical ageing solution.

Fig. S2(A) & Fig. S2(B), ICON scans before and after immersion into the chemical solution.

Supplemental Information 7 Microleakage analysis of teeth surfaces with ICON resin infiltrant.

Stereomicroscope images after immersing into the Methylene Blue dye.

Supplemental Information 8 Micro-Computed Tomography images of ICON resin infiltrants.

Micro-CT images of ICON resin infiltrants showing penetration depth (arrows showing) on the tooth surface.

The authors would like to acknowledge Dr. Mukarram Zubair (College of Engineering, Imam Abdulrahman Bin Faisal University, Dammam, Saudi Arabia) for helping in BET analysis, Intisar Ahmed Siddiqi and Faraz Ahmed Farooqi (College of Dentistry, Imam Abdulrahman Bin Faisal University, Dammam, Saudi Arabia) for statistical analysis.

Additional Information and Declarations

Competing Interests

The authors declare that they have no competing interests.

Author Contributions

Syed Zubairuddin Ahmed performed the experiments, analyzed the data, prepared figures and/or tables, visualization, and approved the final draft.

Abdul Samad Khan conceived and designed the experiments, analyzed the data, authored or reviewed drafts of the article, conceptualization, and approved the final draft.

Maram Alshehri performed the experiments, prepared figures and/or tables, and approved the final draft.

Fatimah Alsebaa performed the experiments, prepared figures and/or tables, and approved the final draft.

Fadak Almutawah performed the experiments, prepared figures and/or tables, and approved the final draft.

Moayad Mohammed Aljeshi performed the experiments, prepared figures and/or tables, and approved the final draft.

Asma Tufail Shah analyzed the data, prepared figures and/or tables, resources, and approved the final draft.

Budi Aslinie Md Sabri conceived and designed the experiments, analyzed the data, authored or reviewed drafts of the article, project Administration, and approved the final draft.

Sultan Akhtar analyzed the data, prepared figures and/or tables, data curation, and approved the final draft.

Mohamed Ibrahim Abu Hassan conceived and designed the experiments, analyzed the data, authored or reviewed drafts of the article, validation, and approved the final draft.

Human Ethics

The following information was supplied relating to ethical approvals (i.e., approving body and any reference numbers):

Scientific Research Ethics Committee, Imam Abdulrahman Bin Faisal University, Dammam, KSA has granted the Ethical approval to carry out the study within its facilities.

Ethics

The following information was supplied relating to ethical approvals (i.e., approving body and any reference numbers):

Imam Abdulrahman Bin Faisal University Institutional Review Board approval to carry out the study within its facilities (Ethical Application Ref: RB-2022-02-155, IRB-2023-02-280).

Data Availability

The following information was supplied regarding data availability:

The data is available at figshare: Ahmed, Syed Zubairuddin (2024). In-vitro comparative thermo-chemical aging and penetration analyses of bioactive glass-based dental resin infiltrates. figshare. Figure. https://doi.org/10.6084/m9.figshare.25567737.v2.

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
