# Peer review of "In-vitro comparative thermo-chemical aging and penetration analyses of bioactive glass-based dental resin infiltrates"

_PeerJ, doi:10.7717/peerj.18831_

## Round 0.1 · original submission · Major Revisions

Based on the referee comments, the manuscript needs a detailed careful revision.

Reviewer 1 ·

Basic reporting

The basic reporting of the study is professional. I don't have any remarks.

Experimental design

Authors have to report what kind of international standards (ISO or ADA or different) have been used for specimens preparation and all tests performance?
What kind of standard has determine the size and number of the specimens?
Furthermore Authors have to extend the list of references and discuss the current literature strongly related to the topic doi:10.17219/dmp/133404, doi:10.17219/dmp/171844, doi:10.17219/dmp/137611.
Authors have to explain each first use of abbreviation within abstract and manuscript body.
Authors have to add a legend of used abbreviation below each table and figure.

Validity of the findings

The validity of the findings is high. I don't have further comments.

Additional comments

I don't have additional comments.

Reviewer 2 ·

Basic reporting

no comment

Experimental design

1. The bioactive glass compositions should be disclosed even tough this is still under patent application. We should know at least what kind of components are containing in the bioactive glasses. In addition, what's the size of the fillers? I know you have used "the concentration (2.5 wt.%)" but this is not sound. The surface area would be more important - so I suggest you should present vol% and densities of all glasses. Otherwise you cannot make sure the contact area of glass to tooth would be the same. If the contact area are not the same, you should further evaluate the remineralization ability per area of glass on all your groups.
2. The mixing procedure is not fully disclosed. Hand mix or machine machine? under vacuum or just a magnetic stir bar mix?
3. please check again the LED curing light - 450 mW/cm2 is low and I could not find this model from woodpecker
4. I would like to know clearly, for how many samples that you have been used for each challenge, i.e. pH, thermocycling and chemical challenge. They should be separated, right?
5. two SEM has been utilized, did you name them all?

Validity of the findings

1. Fig 1 is a bit odd - I do not want to see a figure with adjusted dimension. Please use the original dimension to publish
2. The glass particle size and densities are very critical for the findings validity.
3. I do not understand Table 2 : "Shows statistical significance in comparison of material xxx" normally we just say "The different superscripts indicate the groups are statistically different (p<0.05)." ... please make sure what you are writing.
4. It looks like the glass infiltration would make a higher roughness and lower hardness (even though not statistically difference in some groups) - so, how come I should use glass infiltration techniques? I still think your tooth-glass area contact would allow you can have a better interpretation on your results.

Reviewer 3 ·

Basic reporting

This is an interesting article. Some comments are as follows.

Abstract:
Details on the process of resin infiltration are missing in the Abstract. Hence, it is better to add details on the resin infiltration.
Add conclusion also.

Experimental design

Did the researcher follow the mixing protocol of TEGMA and UDMA previously described or it is formulated by the author? Add reference if followed previous research.
For the sample size calculation, they obtained only 4 per group (total of 16) how the authors calculate it, and software or formula used. Please add details.
The teeth were extracted “mainly from periodontal and orthodontic reasons”. If there are other causes, please add and remove the word ‘mainly’.
For the various mechanical properties testing, did the authors follow ISO standards or previous studies? And add the references.
For the microleakage and penetration depth, it showed a significant difference. It is better to do multiple comparisons among each group rather than only with the control. Statistical consultation should be done with the expert.

Validity of the findings

There are lots of studies studying microleakage and penetration depth. Hence the authors need to add what this study is different from others and what is new in this study.

Discussion and conclusion:
Add limitations and future perspectives of this study.
The conclusion is long. Better to summarize the main points.

---

## Round 0.2 · Minor Revisions

The manuscript needs a further revision.

Reviewer 1 ·

Basic reporting

The manuscript has been correctly revised. I don't have further comments.

Experimental design

The manuscript has been correctly revised. I don't have further comments.

Validity of the findings

The manuscript has been correctly revised. I don't have further comments.

Additional comments

The manuscript has been correctly revised. I don't have further comments.

Reviewer 2 ·

Basic reporting

The M&M is not up to standard.

Experimental design

Unfortunately, the materials and methods part is not acceptable. "whereby the concentration (0.0625 gm wt. %) was optimized. " In your resin matrix TEGDMA/UDMA/CQ/EDBA 1.8:0.603:0.012:0.012 (total 2.42), the filler is 0.0625 (i.e. 2.5 wt%) ? This is very odd. You have found the density of fillers - ok. So now you can calculate the vol% of filler in the matrix. There are many papers mentioning vol% of fillers e.g. doi: 10.2186/jpr.JPR_D_21_00177 . You should also proof the surface chemistry would be the same or different if the loading of these three Bioactive glasses are the same.

Validity of the findings

I want to see the surface chemistry would be correlated to your results.

Reviewer 3 ·

Basic reporting

Abstract:

In the Background, it is only written the aim. It is better to add some background of the research and improve the aim "This study aimed ... based on....".

The Methodology is long, the authors can reduce the method and consider adding in the background.

Experimental design

Method:
Please add about the sample size calculation.

In Table 2, it is better to add the P value <0.0001 instead of absolute 0.

Validity of the findings

-

Additional comments

-

---

## Round 0.3 · Minor Revisions

The manuscript needs a further minor revision as suggested by one referee.

Reviewer 2 ·

Basic reporting

Basic reporting is fine.

Experimental design

The addition of vol % is fine, although the calculation equation is not correct - "vol of solute / vol of solution" ? We calculate using using "mass / density" of BG filler. Further, you have shown the vol % of all your groups are not the same , so how can you guarantee they display under similar condition? Same weight % of BG is not correct. Perhaps you need to find to way to standardize (or specifically, normalize) the results, maybe cross check with the release profile of the glass.

Validity of the findings

normalization of the results are necessary.

Additional comments

nil

Reviewer 3 ·

Basic reporting

The authors have addressed all the comments and improved the manuscript. It can be accpted from my side.

Experimental design

-

Validity of the findings

-

---

## Round 0.4 · accepted · Accept

The manuscript is ready for publication.